# Chemical Fractionations of Lead and Zinc in the Contaminated Soil Amended with the Blended Biochar/Apatite

**DOI:** 10.3390/molecules27228044

**Published:** 2022-11-19

**Authors:** Truong Xuan Vuong, Joseph Stephen, Tu Binh Minh, Thu Thuy Thi Nguyen, Tuan Hung Duong, Dung Thuy Nguyen Pham

**Affiliations:** 1Faculty of Chemistry, TNU-University of Science, Tan Thinh Ward, Thai Nguyen City 24000, Vietnam; 2School of Materials Science and Engineering, University of NSW, Kensington, NSW 2052, Australia; 3Institute of Resources, Ecosystem and Environment of Agriculture, Center of Biochar and Green Agriculture, Nanjing Agricultural University, Nanjing 210095, China; 4School of Environmental and Rural Science, University of New England, Armidale, NSW 2351, Australia; 5ISEM and School of Physics, University of Wollongong, Wollongong, NSW 2522, Australia; 6Faculty of Chemistry, TNU-University of Education, 20 Luong Ngoc Quyen Street, Thai Nguyen City 24000, Vietnam; 7Institute of Chemistry, Vietnam Academy of Science and Technology, 18 Hoang Quoc Viet, Hanoi 10000, Vietnam; 8NTT Institute of Applied Technology and Sustainable Development, Nguyen Tat Thanh University, Ho Chi Minh City 754000, Vietnam; 9Faculty of Environmental and Food Engineering, Nguyen Tat Thanh University, Ho Chi Minh City 754000, Vietnam

**Keywords:** bioavailability, biochar properties, heavy metal remediation, soil amendment, soil remediation

## Abstract

Heavy metal contamination in agricultural land is an alarming issue in Vietnam. It is necessary to develop suitable remediation methods for environmental and farming purposes. The present study investigated the effectiveness of using peanut shell-derived biochar to remediate the two heavy metals Zn and Pb in laboratory soil assays following Tessier’s sequential extraction procedure. The concentration of heavy metals was analyzed using Inductively coupled plasma mass spectrometry (ICP-MS). This study also compared the effectiveness of the blend of biochar and apatite applied and the mere biochar amendment on the chemical fractions of Pb and Zn in the contaminated agricultural soil. Results have shown that the investigated soil was extremely polluted by Pb (3047.8 mg kg^−1^) and Zn (2034.3 mg kg^−1^). In addition, the pH, organic carbon, and electrical conductivity values of amended soil samples increased with the increase in the amendment’s ratios. The distribution of heavy metals in soil samples was in the descending order of carbonate fraction (F2) > residue fraction (F5) > exchangeable fraction (F1) > Fe/Mn oxide fraction (F3) > organic fraction (F4) for Pb and F5  ≈  F2 > F1 > F3 > F4 for Zn. The peanut shell-derived biochar produced at 400 °C and 600 °C amended at a 10% ratio (PB4:10 and PB6:10) could significantly reduce the exchangeable fraction Zn from 424.82 mg kg^−1^ to 277.69 mg kg^−1^ and 302.89 mg kg^−1^, respectively, and Pb from 495.77 mg kg^−1^ to 234.55 mg kg^−1^ and 275.15 mg kg^−1^, respectively, and immobilize them in soil. Amending the biochar and apatite combination increased the soil pH, then produced a highly negative charge on the soil surface and facilitated Pb and Zn adsorption. This study shows that the amendment of biochar and biochar blended with apatite could stabilize Pb and Zn fractions, indicating the potential of these amendments to remediate Pb and Zn in contaminated soil.

## 1. Introduction

Along with the rapid development of industrialization and urbanization, heavy metal contamination has caused a severe threat to the ecological environment and human life [1,2]. Heavy metal contamination in soil has been regarded as a global environmental issue [3]. The primary sources of heavy metal accumulation in soil stem from human activities such as mining operations, smelting, transportation, and farming [4,5,6]. Heavy metals (HMs) which are not fixed in the soil would interact and spread throughout the soil through diverse mechanisms, such as ion exchange, adsorption, precipitation, and complexion [7]. Moreover, heavy metals are both non-biodegradable and can remain in the soil for an extended period [8]. Heavy metals can enter the food chain through crops and accumulate primarily in the human body through diet and digestion [9]. Many heavy metals are toxic and detrimental to human health, even at infinitesimal concentrations [10]. When heavy metals exceed the safety levels in the human body, they can cause harmful effects on human health, such as adverse impacts on the endocrine system, immunity, neurological ailment, and cancer [11,12]. Therefore, it is urgent to search for an effective solution for reducing or remediating heavy metals in contaminated soil, especially in agricultural soil.

Many physical, chemical, and biological methods have been applied to remediate heavy metals in polluted soil [8,13]. Among the current techniques, biochar has been shown to be a potential material to remediate heavy metals in contaminated soil [14,15,16]. Biochar is a material produced by the pyrolysis process of biomass in an oxygen-starved environment [17,18]. Derived from various sources, such as woody and agricultural wastes, biochar is a porous and carbon-enriched material with extraordinary properties, such as a large specific surface area, multiplexed oxygen-containing functional groups, and high adsorption performance [13,19]. Biochar can remediate heavy metals in contaminated soils through various mechanisms, including complexation, reduction, cation exchange, electrostatic attraction, and precipitation reactions [20,21,22].

The remediation efficiency of biochar toward heavy metals depends on many factors such as the original biomass, pyrolysis temperatures, types of heavy metals, soil type, and application rates [23,24]. Previous studies have utilized the biochar produced from biomass, primarily agricultural wastes, including rice straw [25,26], wheat straw [27,28,29], bamboo [30,31], peanut shell [32], and corncob [33,34,35]. These biomasses are cost-effective and available for biochar production. In addition to biochar, inorganic compounds and minerals can be used to immobilize heavy metals as an amendment in soil. Apatite, which is a phosphate mineral, has been reported to remediate heavy metals quite effectively [36,37]. However, it is well-documented that combining multiple biochar or biochar with minerals could yield better results in immobilizing heavy metals in soil, as compared to using biochar or mineral alone [38,39]. Many previous studies have reported that combining biochar with apatite ore is a cost-effective and straightforward method to immobilize heavy metals in soil. Combining rice straw-derived biochar with apatite ore at a ratio of 3:3 (% *w/w*) could significantly decrease Pb, Zn, and Cd in the exchangeable fraction, since the efficiency of heavy metal remediation depends on biochar type, amended ratios, soil type, and heavy metals [40]. Therefore, it is essential to investigate the combination of apatite ore with other types of biochar to thoroughly understand the capability of removing heavy metals in the soil of this combination.

Vietnam has many industrial zones and ore mines, and one of them is the Pb/Zn mine located at the Hich village in Dong Hy district, Thai Nguyen province, Vietnam. Heavy metals have severely contaminated the soil in this area through anthropogenic activities, especially mining operations, which has been reported in previous studies [40,41]. A few studies have focused on using a mixture of biochar/apatite [40] or biochar/fly ash/apatite [42] to remediate heavy metals in the tailing soil. However, the study focusing on agricultural land in this area has remained lacking so far.

Agricultural waste-derived biochars applied to remediate heavy metals in soil have been widely investigated so far [17,21]. Peanut shell is one of the agricultural wastes that is abundant and cost-effective. Several studies have investigated the ability of peanut shell-derived biochar in remediating heavy metals and contaminants in the aqueous solution and soil [41,42,43,44,45,46]. However, there have not many studies investigating the combination of apatite and peanut shells-derived biochar used to remediate heavy metals in the heavy metal contaminated soil so far [40]. In this study, the efficacy of peanut shell biochar produced at different temperatures (400 °C and 600 °C), followed by its combination with apatite in immobilizing Pb and Zn in polluted agricultural soil has been investigated. It is well-documented that most of the previous studies have selected the proportion of biochar in the soil in the range of 3–10% [18]. Nevertheless, for 10% of biochar which is about 100 tons/ha, the cost of biochar would increase exceedingly to $500/ton. While the high ratio is impractical in the field due to the extremely high cost, the lower rate of biochar (1%) is not enough to observe the significant change in the exchangeable fraction of heavy metal in the contaminated soil [40,42]. Therefore, the present study focuses merely on the ratios of 3%, 5%, and 10%.

The present study aims to study agricultural soil and blend it with biochar derived from peanut shells and a combination of biochar with apatite ore to investigate (1) the chemical fractions of Pb and Zn in the agricultural soil; (2) characteristics of peanut shell-derived biochar produced at 400 °C and 600 °C; and (3) Effect of biochar and the mixture of biochar and apatite to the chemical fraction of heavy metals (Pb and Zn), especially the change of the exchangeable fraction for the remediation of heavy metal contaminated soil.

## 2. Materials and Methods

### 2.1. Sample Collection and Amendment Preparation

Surface soil samples (around 0–20 cm depth, 30 cm width, and 30 cm length) were collected in a corn field near the Pb/Zn mine at Hich village, Dong Hy district, Thai Nguyen province (21°43′46.27″ N; 105°51′2.75″ E), Vietnam. Five subsamples which were around 10 kg of total weight and located five meters from each other, were collected, combined, and blended thoroughly to form a homogeneously representative soil sample (CS). Biochar was produced by (1) washing the peanut shells with deionized water and air-dried for 24 h, followed by (2) pyrolyzing at 400 °C (PSB400) and 600 °C (PSB600) using a drum pyrolyser for 2 h [40]. The second amendment is apatite ore purchased from Vietnam Apatite Ltd., located in Lao Cai province, Vietnam (22°29′8.02″ N; 103°58′14.38″ E). The absorbents, biochar and apatite, were ground to less than 1 mm before amending to the soil [40].

### 2.2. Soil Incubation and Experiment Design

The soil sample was homogenized completely and blended with biochar and apatite at various ratios. Briefly, 100 g of soil was placed in a plastic cup and mixed with biochar and apatite at 3%, 5%, and 10%. The experiment was set up in 11 plots, and the characterization of 11 samples is described in Table 1. All incubated soil samples were kept in the shade at room temperature for 30 days. During the incubation period, deionized water was added to the soil samples every two days to maintain the moisture at 70% [40,44].

Then, the soil samples were air-dried at 45 °C for two days and ground to pass through a 2 mm sieve to analyze heavy metal speciation for further analysis.

### 2.3. Analysis Method of Soil and Amendment

The basic properties of the soil samples and amendments (biochar and apatite) were analyzed before and after 1 month of incubation. The pH value of the soil and amendments was determined by a pH meter with a mixture of soil samples and distilled water at a ratio of 1:2.5 [44]. Electrical conductive (EC) was measured by diluting soil and amendments (biochar and apatite) with deionized water (1:1 by volume) using a Hanna HI 9124 Eh/EC meter (Hanna Instruments, Strada Hanna NUSFALAU; Salaj, Rumania) [40]. The pipette method was applied to establish the soil texture and the proportion of clay, silt, and sand in the studied soil [45]. Organic carbon (OC) in soil was measured using the Walkley-Black titration method [46]. The total Pb, Zn, Cd, and As concentration was analyzed using ICP-MS Agilent 7900. A total of 0.1 g of soil was mixed with 8 mL HNO_3_: HCl (*v/v* = 1:3) solution and digested in the microwave system. The operating parameters of the microwave system are shown in Appendix A. The optimal parameters of ICP-MS, and the recovery of Pb, Zn, and Cd were tested using the sediment standard reference material MESS-4 in the range from 92.11 to 109.27% (Appendix A). The chemical fractions of heavy metals were investigated using Tessier’s sequence extraction process. Five fractions included exchangeable (F1), carbonate bound (F2), Mn/Fe-hydroxide (F3), organic substance bound (F4), and residue (F5). Tessier’s sequence extraction detail was given in Appendix A.

X-ray diffraction (XRD) analysis was conducted to identify crystallographic structures in the sample using a computer-controlled X-ray diffractometer (D2 Phraser, Bruker Corporation, 40 Manning Road Billerica, MA, USA) equipped with a stepping motor and graphite crystal monochromator. Surface functional groups on the amendments (PSB400, PSB600 and AP) were characterized using Fourier transform infrared spectroscopy (FTIR, JASCO FT/IR- 4600, JASCO International Co., Ltd., Tokyo, Japan) in the wavenumber range of 400 to 4000 cm^−1^ [40,47]. The surface morphology and chemical composition of amendments were investigated by a field emission electron microscope (FE-SEM, JSM-6700F) equipped with an energy dispersive spectrometer (EDS) [47,48,49]. The Branueur–Emmet–Teller (BET) analysis was applied to investigate the biochar’s particle and dimensional pore characteristics. The biochar’s micropore volume, surface area, and pore width were measured using a BET analyzer (TriStar II 3020, Micromeritics Instrument Corporation One Micromeritics Drive, Norcross, GA, USA).

### 2.4. Statistical Analysis

Data analysis was performed using Excel 2019 and Origin Pro 2021 (OriginLab Corp., Northampton, MA, USA). The difference between the treatments was tested with one-way ANOVA using SPSS 22 (IBM, SPSS Inc., Chicago, IL, USA). The least significant difference (LSD) test was applied to evaluate the differences among the mean values, with *p* < 0.05 considered significant. Principle component analysis (PCA) and Pearson correlation were performed using Origin Pro 2021.

## 3. Results and Discussion

### 3.1. Characteristics of the Investigated Soil and Amendments

The basic physical and chemical properties of the investigated soil sample were analyzed. The results are shown in Table 2. The soil is sandy soil with a high rate of sand (69.78 ± 0.72%), while its clay and silt contents were only 24.74 ± 0.43% and 5.48 ± 0.32%, respectively. The investigated soil had a low OC content with a mean value of 2.49 ± 0.12%, and an EC mean value of 136.5 ± 0.5 µS/cm. The pH value of the soil is 6.47, indicating that the soil type is slightly acidic. The mean concentrations of Pb, Zn, and Cd in the soil sample were 3047.8 ± 98.6 mg kg^−1^, 2034.3 ± 35.4 mg kg^−1^, and 14.1 ± 0.9 mg kg^−1^, respectively. This result was consistent with previous studies [40,42], which reported that the concentration of Pb, Zn, and Cd was primarily in the descending order of Pb > Zn > Cd, and the concentrations of Pb and Zn in the soil samples collected at the same area ranged from about 2000 to 3000 mg kg^−1^. These heavy metal concentrations were significantly higher than the acceptable standard of the ‘National Technical Regulation on the Allowable Limits of Heavy Metals in Vietnamese Soils’ and the United States Environmental Protection Agency (US EPA 2010). According to the US EPA (2010), the permissible limit standard of Pb, Zn, and Cd in agricultural soil was 200, 300, and 3 mg kg^−1^, respectively [50]. According to the Vietnamese Regulation, the acceptable Pb, Zn, and Cd concentrations for agricultural soil are 70, 200, and 1.5 mg kg^−1^, respectively (QCVN 03-MT:2015/BTNMT) [40,51]. The concentrations of Pb, Zn, and Cd in the soil sample were approximately 43, 10, and 9 times higher than the acceptable limits given by the Vietnamese Regulation, indicating that the soil is highly contaminated with these heavy metals. Therefore, it is necessary to develop an effective method to remediate these heavy metals, especially Zn and Pb, due to their intensively high concentration in the studied soil.

Subsequently, the characteristics of biochar pyrolyzed and apatite were investigated. Results have shown that the pH values of PSB400, PSB600, and APA are 10.90 ± 0.01, 11.13 ± 0.02, and 9.36 ± 0.02, respectively (Table 2). These pH values are higher than the soil pH, which shows the possibility of increasing the pH of the soil after incubation. The high pH values of biochar could be attributed to the loss of acidic groups (-COOH and -OH), the formation of carbonate and the alkali salts separating from organic compounds when the pyrolysis temperature increased [50,51,52]. The OC values of biochar and apatite are shown in Table 2. Both investigated types of biochar are rich in carbon with 80.79 ± 8.34% and 73.34 ± 0.21% for PSB400 and PSB600, respectively, while the apatite is poor with OC (3.34 ± 0.21%). In addition, the EC values of PSB400, PSB600, and apatite are >1900, 1104.51 ± 1.50, and 1104.53 ± 1.51 µS cm^−1^, respectively, which are significantly higher than that of the soil (118.50 ± 0.50 µS cm^−1^). This indicates good electrical conductivity of biochar and apatite.

The contents of Pb and Cd in PSB400, PSB600, and apatite were below the detection limit. The mean concentration values of Zn in PSB400, PSB600, and apatite are 0.70 ± 0.03, 9.43 ± 0.03, and 9.43 ± 0.03 mg kg^−1^, respectively (Table 2), illustrating that these materials are suitable for soil amendments to remediate heavy metals in soils.

### 3.2. Analysis of the Amendments’ Characteristics 

#### 3.2.1. XRD Analysis

XRD is applied to investigate the structural properties of apatite ore, PS, PSB400, and PSB600 using monochromatic CuKα radiation (λ = 1.54056 Å) operated at 40 kV (tube voltage) and 40 mA (tube current) (Figure 1). Figure 1A shows the XRD patterns of PS, PSB400, and PSB600. The XRD results show both amorphous and crystalline structures, and the presence of some inorganic minerals, such as SiO_2,_ in the PS, with the typical peaks at 2θ = 21.6° and 26.6° [53]. The peaks at 2θ in the range of 20–25° in the PS sample are assigned to the crystalline structure of the cellulose contained in the raw biomass. These peaks are the most pronounced in PS, followed by PSB600 and PSB400. All three materials have the same peak (002) at 2θ = 26.3° which is attributed to the graphite content [54], indicating the stacking of the graphite basal planes of carbon crystallite structures [55]. Another plane (100) was located at 2θ = 43.5° in PS and PSB600, and at 2θ = 44.1° in PSB400 was assigned to graphitic content [56], illustrating that the graphite-like atoms were ordered in a single plane [55]. 

From XRD results, these materials exhibit the dominant diffraction peak at around 2θ = 20°–30°, illustrating an amorphous structure that was disorderly stacked by carbon rings [57]. Overall, the XRD results show that the XRD patterns of PS, PSB400, and PSB600 are identical to the peaks for the phases of graphite and carbon. The XRD peaks in PS are resulted from the amorphous and crystalline structures of cellulose and lignin, while those in PSB400 and PSB600 are attributed to the products of decomposed cellulose and lignin [55].

Figure 1B shows the XRD pattern of apatite ore. Most of the peaks of apatite ore are identical to the peaks for the phase of fluorapatite (COD: 96-900-1346/[PDF00-015-0876, 32]) and SiO_2_ (COD: 96-900-1494/[PDF 00-046-1045]), illustrating that the phase composition of the apatite ore is primarily fluorapatite and quartz. This result is in close agreement with the previous study [58].

#### 3.2.2. FT-IR Analysis of Amendments

Figure 2A shows the FTIR spectra of the raw peanut shell (PS), biochar-derived from peanut shell pyrolysed at 400 °C (PSB400), and biochar-derived from peanut shell pyrolysed at 600 °C (PSB600). All the spectra of PS, PSB400, and PSB600 have a prominent peak at ~3418 cm^−1^, which can be attributed to the hydroxyl group (-OH) [59,60]. The intensity of this peak is the most pronounced in PS and more negligible in PSB400 and PSB600 when the pyrolysis temperature increases. Additionally, the spectra showed a peak at around 2925 cm^−1^ which is the most evident in PS, followed by PSB400 and almost flat in PSB600, which can be associated with C-H [61,62]. The spectra of three materials have another peak at around 2374 cm^−1^ which can be assigned to the C≡C stretch alkynes functional group [57]. This peak is most pronounced in PS, weaker in PSB400 and intensively negligible in PSB600. The peak at around 1744 cm^−1^ refers to the stretch vibration of aromatic C = O functional group, carboxylic groups, or conjugated ketone [62,63]. The C=O stretching and aromatic C=C vibrations were attributed to the peak at around 1510–1580 cm^−1^ [62,64], while those at 1300–1340 cm^−1^ and 1270 cm^−1^ can be associated with phenolic -O-H and C-O, respectively [65].

The peak at around 1034 cm^−1^ refers to C-O-H or C-O-C stretching or allopathic [59,66]. This peak is prominent in PS, while being negligible in PSB400 and PSB600. A peak at 563–600 cm^−1^ which was pronounced in PS and negligible in PSB400 and PSB600, can be attributed to a C-O-H out-of-plane bending mode of aromatic compounds [63]. After heating at 400 °C and 600 °C, O-H, the aliphatic C-H and C-O stretching vibrations at 3418, 2925, and 1034 cm^−1^ decreased significantly, illustrating the decomposition of cellulose, hemicellulose, and lignin contents [67]. According to the FT-IR results of PS, PSB400, and PSB600, all absorption bands are mainly attributed to the hydroxyl group in cellulose, carbonyl groups of acetyl ester in hemicellulose, and carbonyl aldehyde in lignin [57].

For the apatite ore, Figure 2B shows that there are four characterized peaks at 1904, 1049, 574, and 464 cm^−1^ in the FT-IR spectra of the apatite ore, which represent the characteristic bands of PO_4_^3-^ and are associated with asymmetric stretching vibration of the P-O bond and the asymmetric bending vibration of O-P-O, respectively [58,68]. The peaks at 3618 and 3446 cm^−1^ are pronounced and attributed to the vibration of the OH group or the absorbed water [58,69]. The peak at 1619 cm^−1^ is assigned to the bending vibration of the O-H group [58]. The peak at around 797 cm^−1^ corresponds to the presence of the F^−^ ion. In addition, the presence of CO_3_^2−^ stretching vibration at 1424-1461 cm^−1^ confirms that the sample is a fluor-hydroxide-carbonate-apatite [69,70]. This FT-IR result agrees with the result from previous studies [58,69,70]. The FT-IR result shows that the apatite ore has some main functional groups such as PO_4_^3−^, OH^−^, or CO_3_^2−^ that can combine with the heavy metals by precipitation or exchange reactions.

#### 3.2.3. SEM and EDS Analysis of Amendments

SEM and EDS analysis were performed to investigate the microstructure and compositions of the studied materials. SEM micrographs for the external morphology of PSB400, PSB600, and APA are shown in Figure 3. Specifically, Figure 3A,B illustrate that the exterior surface of PSB400 is non-uniform and heterogeneous and the PSB600 has a more porous surface structure than PSB400. In contrast, the APA’s surface is smooth and has a compact construction with no visible holes (Figure 3C). The SEM images reveal that PSB600 has a higher content of macro pores than PSB400 and APA, indicating that it can adsorb a higher content of heavy metals than PSB400 and APA. 

Furthermore, EDS was applied to investigate the composition of the materials’ surfaces. The EDS graphs and the information on the elements of the external surfaces of materials are demonstrated in Figure 3. Figure 3 illustrated that C and O were the primary elements that contribute to the composition of PSB400 and PSB600, while trace amounts of elements such as Ca, K, Fe, Ca, Si, Mg, and Al were detected in the samples. The apatite ore had a higher percentage of O, Si, Ca, F, and P than in biochar samples, indicating that the apatite ore was created from those elements. This result agreed with the outcome of XRD analysis which shows that the primary components of apatite ore might be speculated as fluorapatite and quartz.

Results from BET analysis have shown that the surface area of PSB600 (79.6294 m² g^−1^**)** is much higher than that of PSB400 (1.4787 m² g^−1^) and APA (0.4487 m² g^−1^) (Table 1), proving the effects of the temperature on carbonization. At 600 °C, most of the oxygen and hydrogen was released as gases (CH_4_, H_2_, CO), thus the carbon was concentrated. Therefore, PSB600 has a greater porosity and surface area to adsorb and bind to heavy metals than PSB400.

Overall, the PSB600 had a larger surface area and porosity than PSB400 and APA, yet fewer organic functional groups than PSB400. The reason for the larger surface area of PSB600 than PSB400 was attributed to water losses in the dehydration process and the release of the volatiles (CH_4_, H_2_, and CO) from the biomass, particularly through the decomposition of benzene-like aromatic-containing compounds in the lignin [71,72,73]. In addition, the higher the temperature is, the more organic functional groups in biochar are degraded [74]. Therefore, PSB400 is richer in functional groups than PSB600. The difference in the number of organic function groups of two kinds of biochar was pronounced in the FT-IR spectra (Figure 2A). This result indicated that the mechanisms by which the two treatments adsorb and bind to the heavy metals would differ.

### 3.3. Effects of Biochar and Apatite Amendment on OC, pH, and EC after a 30-Day Incubation

#### 3.3.1. pH

The soil pH plays a vital role in remediating the heavy metals in soil and correlates significantly with the exchangeable fraction of heavy metals and their bioavailability in the soil. Table 3 shows that the pH values of all biochar-spiked samples were significantly higher than the CS sample. Furthermore, the soil pH slightly increased upon the addition of PSB400, PSB600, and APA. The high ratio of biochar amended resulted in high pH values in the spiked soil samples. The pH reached the highest values in the samples PB4:10 (pH = 7.24 ± 0.01) and PB6:10 (pH = 7.23 ± 0.01) when being treated with PSB400 and PSB600 biochar (10%). These results agree with previous studies [40,42] and are attributed to the high pH values of biochar and apatite ore that are significantly greater than that of CS. The high pH values of biochar and apatite ore might be vital in reducing the exchangeable fraction of Pb and Zn in the studied soil samples.

#### 3.3.2. Organic Carbon (OC)

Table 3 illustrated the OC in soil samples before and after incubating with PSB400, PSB600, and apatite ore for 30 days. In contrast to apatite, biochar is a rich carbon material. Therefore, all biochar spiked soil samples had a significantly higher OC value than CS, and the OC values in soil samples increased proportionally with the amended biochar ratios. The highest OC value after incubation (86.46 ± 0.80 g kg^−1^) was observed in PB4:10 spiked with 10% PSB400 biochar, which is slightly higher than PB6:10 spiked with the same 10% proportion of PSB600 biochar. Table 3 shows that the OC values of soil samples spiked with PSB400 were relatively higher than that of soil samples spiked with PSB600, indicating that PSB400 contributed more significantly to the OC values of soil samples than PSB600. This can be explained by the quantity of OC that was decomposed more significantly at 600 °C than at 400 °C. Therefore, the PSB400 is richer in OC than PSB600. Moreover, as PSB400 has more organic function groups than PSB600, it has more binding sites than PSB600 which causes higher OC in soil samples spiked by PSB400 than PSB600. This result agrees with previous studies where the increase of OC in biochar-treated soil was observed upon increasing rates of biochar application [20,71,72]. 

#### 3.3.3. Electrical Conductivity (EC)

Electrical conductivity is a valuable index for studying soil heavy metal remediation since it can be used to explain variability in the physical and chemical properties of the soil [15]. Basically, a solution conducts an electrical current well when it has a high EC value (or a high salt concentration). As shown in Table 3, the EC value of the CS was significantly different from all amended soil samples. The sample PB6:10 had the highest EC value (231.3 ± 2.1 µS cm^−1^), which was higher than that of the sample PB4:10 (211.3 ± 2.3 µS cm^−1^). Table 3 shows that the EC values of the soil samples spiked with PSB400 were relatively lower than those spiked with PSB600. It can be clarified that the biochars produced at high temperatures typically have higher EC values than those produced at lower temperatures [74,75]. Overall, the EC values of soil samples increased when the biochar and apatite amended ratios increased. This result is consistent with previous studies which reported that the EC value increased with the increasing biochar applications [2,71]. 

### 3.4. Effects of Amendments on the Pb and Zn’s Chemical Fractionations in Soil Samples

The data information about the chemical fractions of Pb and Zn (mg kg^−1^) is shown in Appendix A. The proportions of chemical fractions of metals after 30 days of incubation with biochar and apatite ore are described in Figure 4 and Appendix A. 

#### 3.4.1. Chemical Fraction of Pb

*Exchangeable fraction:*Appendix A and Figure 4A show that CS had the Pb concentration distributed in the descendent order of F2 > F5 > F1 > F3 > F4 with the percentage of 63.66, 17.99, 16.67, 1.06, and 0.62, respectively. The highest and lowest Pb concentrations were obtained in the carbonate fraction (F2) and organic carbon fraction (F4), respectively. After incubation, the chemical fractions of Pb were varied with the different amendments and amended ratios. The Pb’s exchangeable fraction in amended samples ranged from 8.47 ± 0.66% in the PB6:10 sample to 11.60 ±1.45% in the PB6:3 sample. All the spiked samples had a significantly lower exchangeable fraction of Pb than that of CS (16.67 ± 0.17%) (*p* < 0.05). This result demonstrated that adding biochar and the biochar/apatite mixture in the soil samples would decrease the F1 of Pb. The high amended ratios reduced the exchangeable fraction of Pb. Specifically, the Pb’s exchangeable fraction (F1) decreased when the applied ratios of biochar increased from 3% to 10%, and PSB600 was slightly more effective in reducing the exchangeable fraction of Pb than PSB400. The blend of biochar and apatite could significantly reduce Pb in the exchangeable fraction but relatively lower than solely amended biochar. This result was consistent with previous studies which reported a significant decrease in the Pb’s portion in the exchangeable fraction [40,76].

*Carbonate bound fraction:*Appendix A and Figure 4A illustrated that Pb dominated the carbonate fraction, with approximately two-thirds of all fractions. The proportion of F2 in the CS was 63.66%, while the other amended samples had higher F2 values than CS, yet there was no significant difference between them. Only the sample PB4:5 had a significantly different F2 compared to CS. These results indicated that adding biochar and apatite to the soil samples did not change the carbonate fraction, except for the sample PB4:5, which contains the mixture of biochar and apatite at a ratio of 5%. There was no difference between the two types of biochar (PSB400 and PSB600) in F2 proportion when being spiked in soil samples.

*Fe/Mn oxide bound fraction:* The proportion of Pb in the F3 was minimal and ranked fourth, after F2, F5, and F1. The F3 ratio in the unamended and amended soils was in the range of 0.90% to 1.22%. The proportions of Pb in this fraction of all amended soil samples were slightly higher than that of CS, but there was no significant difference, except for the samples PB4:10 and PB6:10, which contained biochars at a 10% ratio (Appendix A). This result shows that at the highest proportion of biochar (10%), the Fe/Mn oxide bound fraction of Pb was slightly increased compared to that of CS and remained unchanged in other investigated ratios and mixtures. 

*Organic matter bound fraction:* This fraction had a minuscule Pb proportion with merely 0.62% for CS and 0.68 to 2.79% for other amended soil samples. The F4 proportion of Pb in samples PB4:10, PB6A3, and PB6A5 was higher than that of CS, while other samples were either the same (*p* > 0.05) or slightly lower than CS (Appendix A). The increasing proportion of this fraction could be attributed to the addition of biochar in soil samples which provided organic functional groups in the soil. 

*Residual fraction:* Pb in the residual fraction of the CS sample accounted for 17.99% and ranged from 18.78 to 21.44%. This ratio increased proportionally with the increasing ratio of biochar and apatite compared to that of CS. There was a significant increase in F5 in almost amended soil samples, aside from samples PB4A:3, PB4A:5, PB6:3, and PB6:5. These samples did not vary significantly from CS in the F5 fraction of Pb. These results indicated that the increase of Pb in F5 might be at the expense of Pb in F1. The increase in Pb’s portion in the residual fraction in the soil after amending biochar with suitable ratios was also reported in previous studies [40,77].

#### 3.4.2. Chemical Fraction of Zn

*Exchangeable fraction:*Appendix A and Figure 4B show that the exchangeable fraction of Zn in the CS sample accounted for 20.89%, while the amended samples ranged from 14.53% to 21.69%. There was no significant difference in this fraction in PB4:3, PB4:5, and PB6:3 samples, while other amended samples showed a considerable decrease, as compared to CS. This result illustrated that amending biochar at the rate of 10% of PSB400, 5% and 10% of PSB600 or the mixtures of 3:3% and 5:5% of biochar and apatite could decrease the Zn’ portion in the exchangeable fraction or immobilize it in the soil. Results from the present study were similar to the previous study, which reported that amending biochar could significantly reduce the proportion of Zn’s exchangeable fraction [78]. 

*Carbonate-bound fraction:*Appendix A shows that the proportion of Zn in the carbonate-bound fraction was 40.37 ± 2.21% and dominated other fractions. The Zn’s portion in this fraction in amended samples was relatively similar to that of the CS sample. However, there was no significant difference between the unamended and amended samples (*p* > 0.05). This result illustrated that after one-month incubation of biochar PSB400, PSB600, and apatite ore, there was no significant change of Zn bound to the carbonate fraction. This result is consistent with the previous study which also obtained no significant change in the carbonate-bound fraction after amending with biochar [79]. 

*Fe/Mn oxide bound fraction:* For this fraction, there was almost no significant difference between the amended soil samples compared to the CS sample, aside from PB4:5, PB6:3, PB6:5, and PB6:10 samples, which are slightly higher than that of CS. The addition of the blend of biochar and apatite did not induce any significant changes in Zn’s portion in this fraction.

*Organic matter bound fraction:* The mean proportion of Zn in this fraction of CS was 0.46 ± 0.06% and was the smallest in all fractions. After one month of incubation, the percentage of Zn in this fraction in amended samples was significantly different (*p* < 0.05) and slightly increased compared to that of CS, indicating that the addition of biochar contributed more organic substances bonding to Zn. These results are similar to previous studies [40], [80], which also showed the increase of Zn in the organic-bound fraction after incubating with biochar, compared to the CS. Moreover, the addition of PSB400 to the soil samples increased the Zn’s F4 slightly higher than that of PSB600. The highest proportion of F4 for Zn (3.03 ± 0.05%) was present in the PB4:10 sample which was about 6.8 folds higher than that of CS. The higher proportion of Zn in PSB400 compared to PSB600 could be due to the higher concentration of functionalized organic molecules on the surface of the carbon matrix.

*Residual fraction:* The proportion of Zn in the residual fraction of the CS was 35.03 ± 1.05% and ranked second after the carbonate-bound fraction. The F5 fraction of Zn in the amended samples ranged from 31.77 ± 0.82% to 40.82 ± 2.41% (Figure 4B), which was slightly altered compared to the CS. The highest value of Zn in this fraction was the PB4:10 sample (40.82 ± 2.41%), which was about 7% higher than that of CS. There was no significant difference between the spiked samples and CS (*p* > 0.05), except for samples PB4:10, PB6:5, PB6A3, and PB6A5 (*p* < 0.05), which increased slightly higher than that of the CS. A similar result was also reported in the previous study [76], in which a significant increase in the residual fraction of Zn was obtained upon applying biochar amendment in soil with suitable ratios.

The significant effects on the exchangeable fraction of Pb and Zn by the combination of biochar and apatite were observed after 30 days of incubation. For Pb, the exchangeable fraction of control soil was 495.77 ± 5.20 mg kg^−1^ (Appendix A), which decreased slightly to 455.73 ± 15.24 and 328.46 ± 7.32 mg kg^−1^ after being incubated with PSB400 at the ratio of 3% and 5% respectively. However, after being incubated with the combination of 3% of PSB400 and 3% of APA (sample PB4A3) and 5% of PSB400 and 5% of APA (sample PB4A5), the F1_Pb decreased substantially to 324.88 ± 16.60 mg kg^−1^ and 275.37 ± 9.01 mg kg^−1^, respectively. Similarly, after incubating with PSB600 at the ratio of 3% and 5%, the F1_Pb were 319.82 ± 39.99 mg kg^−1^ and 295.26 ± 12.05 mg kg^−1^ and lower than that of the control soil (sample CS). Nonetheless, when the mixture of 3% of PSB400 and 3% of APA (sample PB6A3) and 5% of PSB400 and 5% of APA (sample PB6A5) was applied, the F1_Pb in the samples PB6A3 and PB6A5 reduced significantly to 256.83 ± 28.50 and 252.83 ± 17.81, respectively. These results illustrated that the presence of APA could reduce the exchangeable fraction of Pb (F1_Pb). The reduction of F1_Pb caused by the presence of APA might be attributed to the high concentration of phosphate in APA which upon its reaction with Pb^2+^ lead to the precipitation of [Pb_10_(PO_4_)_3_(CO_3_)_3_FOH] and/or Pb_5_ (PO_4_)_3_ (Cl, OH, F) [81,82,83]. 

For Zn, the alteration of the exchangeable fraction was significantly changed after being incubated with the mixture of biochar and apatite, but the changed level was less than Pb. Appendix A shows that the F1_Zn in the control soil was 424.82 ± 4.69 mg kg^−1^ and changed slightly to 416.97 ± 6.93 and 348.97 ± 34.08 after being incubated with PSB400 at the ratio of 3% and 5%. However, this figure changed substantially when applied to the mixture of 3% PSB400 + 3% APA and 5% PSB400 + 5% APA, which were 326.84 ± 23.90 mg kg^−1^ and 279.72 ± 32.17 mg kg^−1^. This result indicated that APA had a significant influence on decreasing the exchangeable fraction of Zn. Nonetheless, the effect of APA on F1-Zn seems less than that on F1_Pb. Previous studies reported that the sorption of Pb by apatite in solution was better than Zn, due to the precipitation of Pb with phosphate in apatite [83,84,85]. 

These significant changes in F1_Pb and F1-Zn after being incubated with biochar and a mixture of biochar with apatite were reported in the previous study [40]. Dang et al. reported that after 3 months of incubation with rice straw-derived biochar produced at 350 °C at the ratio of 1%, 3%, 5% and the combination of biochar with apatite at the ratio of 3% BC+ 3% APA, the exchangeable fraction of Pb and Zn were substantially decreased from 466 mg kg^−1^ to 300, 194, 146, and 145 mg kg^−1^, and from 506 mg kg^−1^ to 377, 233, 201, and 249 mg kg^−1^, respectively [40]. The slight difference between the treatment effects of this study and Dang’s study might be attributed to the difference in the investigated soil sample, the incubated time and the biochar. 

Biochar amendments affect the mobility and bioavailability of heavy metals in soil, depending on the specific metal, soil type, and soil characteristics such as EC, OC, and pH [2]. The impact of biochar on Pb and Zn availability can be attributed to the differences in the soil type, biochar feedstock, pyrolysis conditions, biochar composition, and biochar application rate [80]. The primary mechanisms of metal immobilization by biochar in soils were generally dependent on the increase in soil pH, ion exchange, adsorption, and precipitation processes, including specific adsorption and precipitation as oxy-hydroxides, with carbonate or phosphate [72].

The results of previous studies also showed that biochar and the combination of biochar and apatite at suitable ratios promoted the transformation of exchangeable heavy metals to relatively stable forms [40,42]. The chemical fractions of Pb and Zn in soils varied when being amended with various biochars, as reported by different studies. It might be attributed to the diversity of the soil environment, biochar characteristics, amended ratios, and incubated time [79].

### 3.5. Correlation of the Exchangeable Fraction of Heavy Metals with pH, OC, and EC

Since the Pb and Zn exchangeable fractions changed the most substantially compared to other chemical fractions with the change of biochar and apatite rates, it is necessary to focus on the correlation of the exchangeable fraction with pH, OC, and EC of the soil samples after incubation.

The Spearman correlation was applied to confirm the relationship between pH, OC, and EC and the exchangeable fraction of Zn (F1_Zn) and Pb (F1_Pb). The correlation results of F1_Pb and F1_Zn with pH, OC, and EC are described in Figure 5. Figure 5A shows that F1_Pb negatively correlated with pH, OC, and EC. The correlation of F1_Pb with pH was the most substantially negative (r = −0.85, *p* < 0.05), while its correlations with OC and EC were relatively moderate with r = -0.6 and r = −0.56 (*p* < 0.05), respectively. This means that as the pH values of the soil samples increased, the Pb’s exchangeable fraction was lowered. Moreover, the correlation between the pH with EC and OC was positively strong (r = + 0.79) and moderate (r = +0.69), and the correlation between OC and EC was strongly positive.

Similarly, as shown in Figure 5B, the exchangeable fraction of Zn had a significantly negative correlation with pH (r = −0.94), OC (r = −0.73), and EC (r = −0.74), while pH, OC, and EC had strongly positive correlations with the r = +0.69, +0.79, and +0.85 for pH and OC, pH and EC, and EC and OC, respectively. The strongly negative correlations of the exchangeable fractions of Pb and Zn were also reported in previous studies where biochar and apatite were amended in the Pb and Zn-polluted soil [40]. 

The negative correlation between pH and the exchangeable fraction of Pb and Zn was attributed to the reason that increasing solution pH leads to a rapid increase in net negative surface charge that may explain the enhanced affinity for metal ions causing the decrease of F1_Pb and F1_Zn in the soil chemical fraction. In addition, the positive correlation between pH and OC in soil samples could be assigned to the high pH value and OC content of biochar. When the biochars, which have high pH values and are rich in OC, are blended in the soil samples. The pH and OC of amended soil samples increased with the increasing amended ratios of biochars. Therefore, OC and pH values of soil samples have a positive correlation as aforementioned. Moreover, biochars have minerals and organic functional groups, which facilitate the higher value of electrical conductivity of the soil. Therefore, OC, EC and pH values in the investigated soils increased with the added ratios of biochars and these factors (OC, EC, and pH) have a positive correlation with each other. The higher values of OC and EC support strongly the combination of heavy metal ions with organic matter and other substances in the solution that causes the reduction of the exchangeable fraction of Pb and Zn in soil samples. Therefore, F1_Pb and F1_Zn had a negative correlation with OC and EC and pH.

Overall, F1_Pb and F1_Zn had strongly negative correlations with pH, EC, and OC, and the correlations between pH and OC, pH and EC, OC and EC were strongly positive, indicating that some changes in pH, EC, and OC might affect each other and the exchangeable fractions of Zn and Pb significantly in the negative direction.

### 3.6. PCA Analysis of the Chemical Fractions of Zinc and Lead with pH, OC, and EC

In addition to the Spearman correlation, performing the principal component analysis of Pb and Zn chemical fractions (F1, F2, F3, F4, and F5) with pH, EC and OC is also essential to understand the correlation of various factors. The information on the PCA analysis is shown in Table 4 and Figure 6. The PCA analysis indicated that the samples of various treatments were distributed in different regions of the data space (Figure 6).

For Pb, the first principal component (PC1) accounted for 48.77% of the total variation, and the second component (PC2) explained 72.27% of the total variance (Table 4), indicating that the presence of the soil amendment agents resulted in the difference. Specifically, the OC, EC, pH, and F1 are the main elements that dominantly contribute to PC1, with r = +0.47, +0.45, +0.48, and −0.48, respectively. The OC, EC, and pH had a moderately positive association, while F1 had a moderately negative association with those elements (pH, EC, OC). These results are in agreement with the results of the correlation aforementioned. In addition, PC1 also showed a weak association between F2, F3, F4, and F5 (r < ±0.25), which had a low loading on this component. In contrast, PC2 was dominated by F3, F4, and F5, which are associated moderately with PC2 with r = +0.39, 0.58, and +0.58, respectively. Meanwhile, F2 is moderately negatively associated with PC2 (r = −0.58). These results illustrate that pH, OC, and EC had a significant effect on F1 and slightly affected F3, F4, and F5. In addition, F3, F4, and F5 were moderately affected by PC2 but in an opposite direction as F2. Figure 6A shows that the locations of the amended treatments were varied and grouped in the regions where the treatments with similar effects on heavy metals fractions were grouped. They are CS, PB4:3, PB4:5, and PB6:3 (group 1); PB4A3, PBA5, and PB6:5 (group 2); and PB4:10, PB6:10, PB6A3, and PB6A5 (group 3); in which group 1 had no or the least effect on the change of F1_Pb, and the group 3 induced the most substantial changes of the F1_Pb.

For Zn, PC1, and PC2 accounted for 60.10% and 16.11%, respectively, of the total variation (Table 4). PC1 was dominated by F1, pH, OC and EC, which have a moderate loading on this component (PC1) with r = −0.43, + 045, + 0.40, and + 0.41, respectively. These results are consistent with the results of correlation as well. Additionally, Table 4 shows that F2, F3, F4, and F5 had a low loading on PC1 with r < ±0.3. However, PC2 was dominated by F2, F3, and OC, which had an intermediate loading on this component with r = −0.57, +0.66, and +0.32, respectively. F1, F4, F5, pH, and EC had a minimal contribution to PC2, indicating that the changes in these factors did not significantly affect F2, F3, and OC. In addition, Figure 6B indicates that the locations of treatments varied in different areas of the data space. They might be classified into four groups where the treatments with closely similar effects on F1_Zn were gathered in the same region, such as Cs, PB4:3 (group 1); PB4:5 and PB6:3 (group 2); PB4A3, PB6:5, and PB6:10 (group 3); PB6A3, PB6A5, PB4:10, and PB4A5 (group 4).

## 4. Conclusions

Mining activities have the potential of inducing severe environmental contamination. The Tessier extraction procedure employed in the current study suggests that Zn in the studied soil sample was dominantly associated with the residual fraction (F5) and carbonate fraction (F2), while Pb was primarily associated with the carbonate fraction (F2). The highly significant negative correlations between Pb and Zn’s exchangeable fractions of pH, EC, and OC indicate that biochar and the combination of biochar and apatite had a significant impact on the exchangeable fractions of Pb and Zn in soil, especially at the high amended ratio of 5% and 10% of biochar and biochar/apatite. The present study unveiled that the biochar could turn Pb and Zn from labile into more stable fractions in the amended soil and effectively elevate soil pH, OC, and EC values to reduce Pb and Zn bioavailability. These results promote the effectiveness of biochar application for decreasing heavy metal bioavailability and stabilizing heavy metal in contaminated farmland soil. The current study infers that amending biochar derived from peanut shells and apatite has the potential to remediate heavy metals in contaminated soil. The present laboratory findings necessitate further assessments and validation for future large-scale site-specific applications, especially at farmlands.

## Figures and Tables

**Figure 1 molecules-27-08044-f001:**
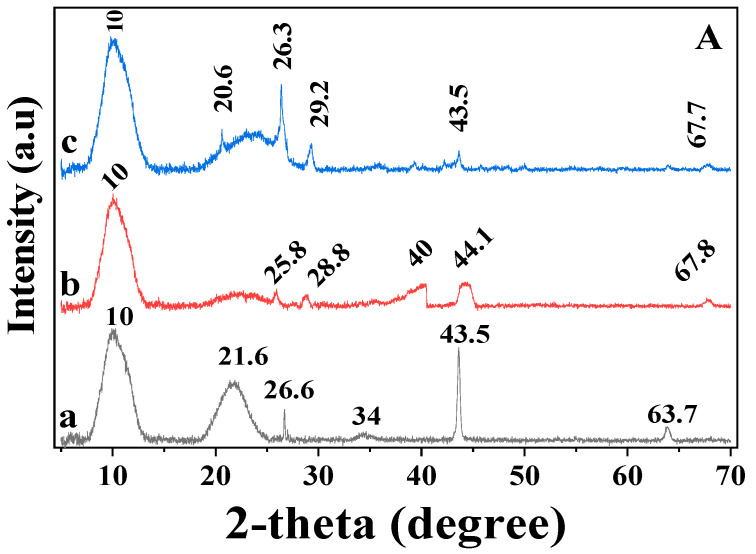
(**A**) XRD spectra of peanut shell (PS) (a); PSB400 (b); PSB600 (c); and XRD spectra **(B)** of apatite ore (APA) compared to XRD pattern of quartz and fluorapatite.

**Figure 2 molecules-27-08044-f002:**
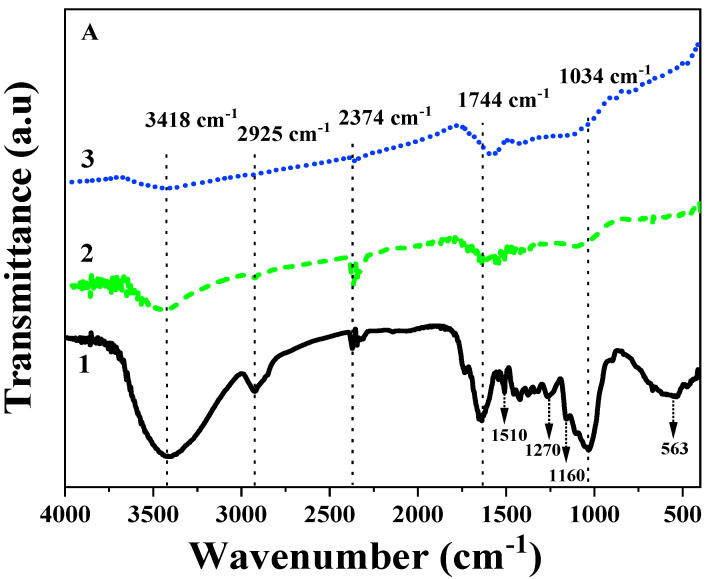
FT-IR spectra of (**A**) peanut shell and its biochar: PS (1), PSB400 (2), PSB600 (3), and (**B**) apatite ore.

**Figure 3 molecules-27-08044-f003:**
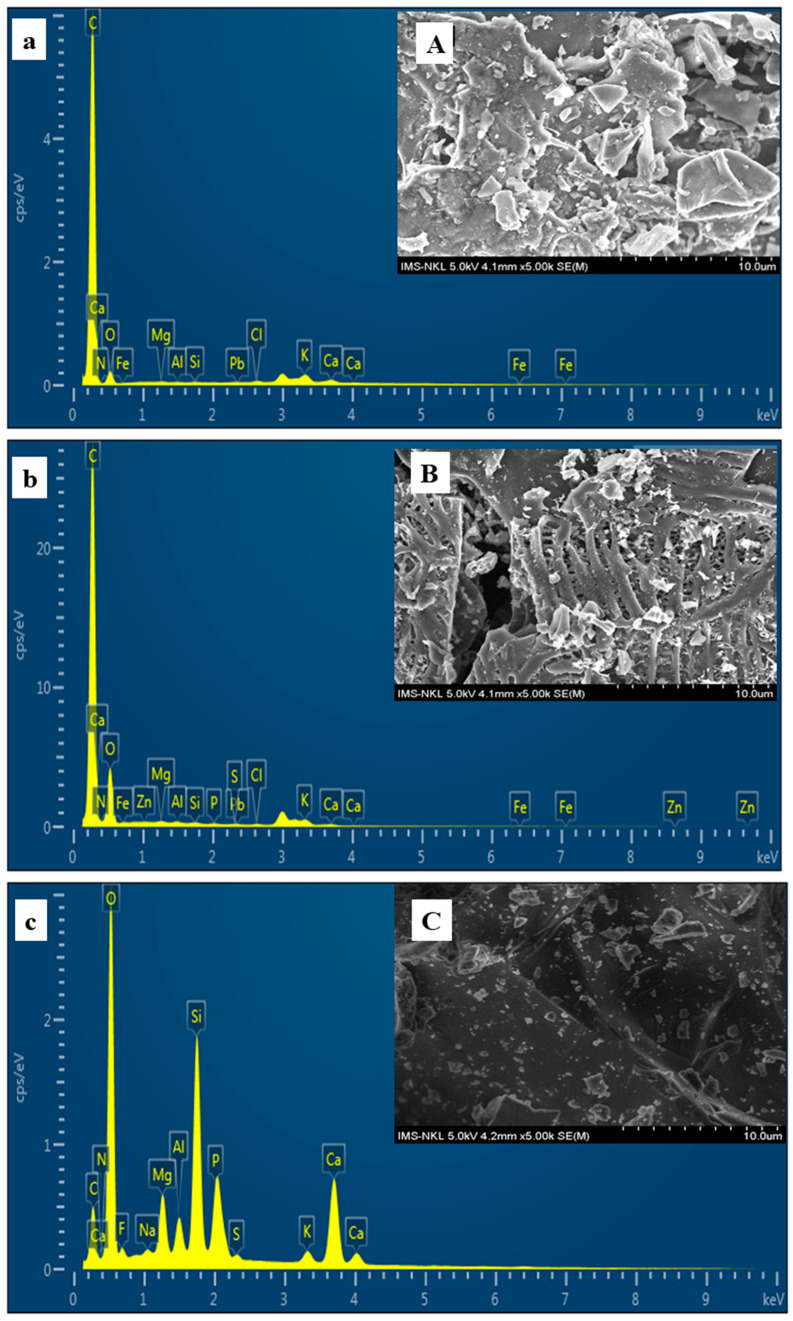
SEM images of PSB400 (**A**); PSB600 (**B**); APA (**C**); and EDS analysis of PSB400 (**a**); PSB600 (**b**); APA (**c**).

**Figure 4 molecules-27-08044-f004:**
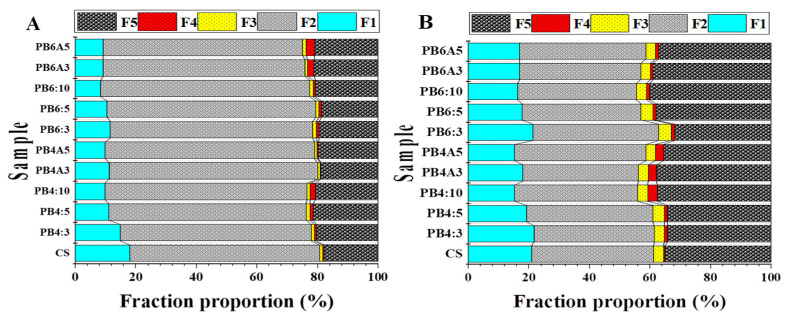
Proportions of chemical fractionations of lead (**A**) and zinc (**B**) in the soil samples after 30-day incubation (F1: exchangeable; F2: carbonate bound; F3: Fe/Mn- hydroxide; F4: organic matter bound; F5: residue).

**Figure 5 molecules-27-08044-f005:**
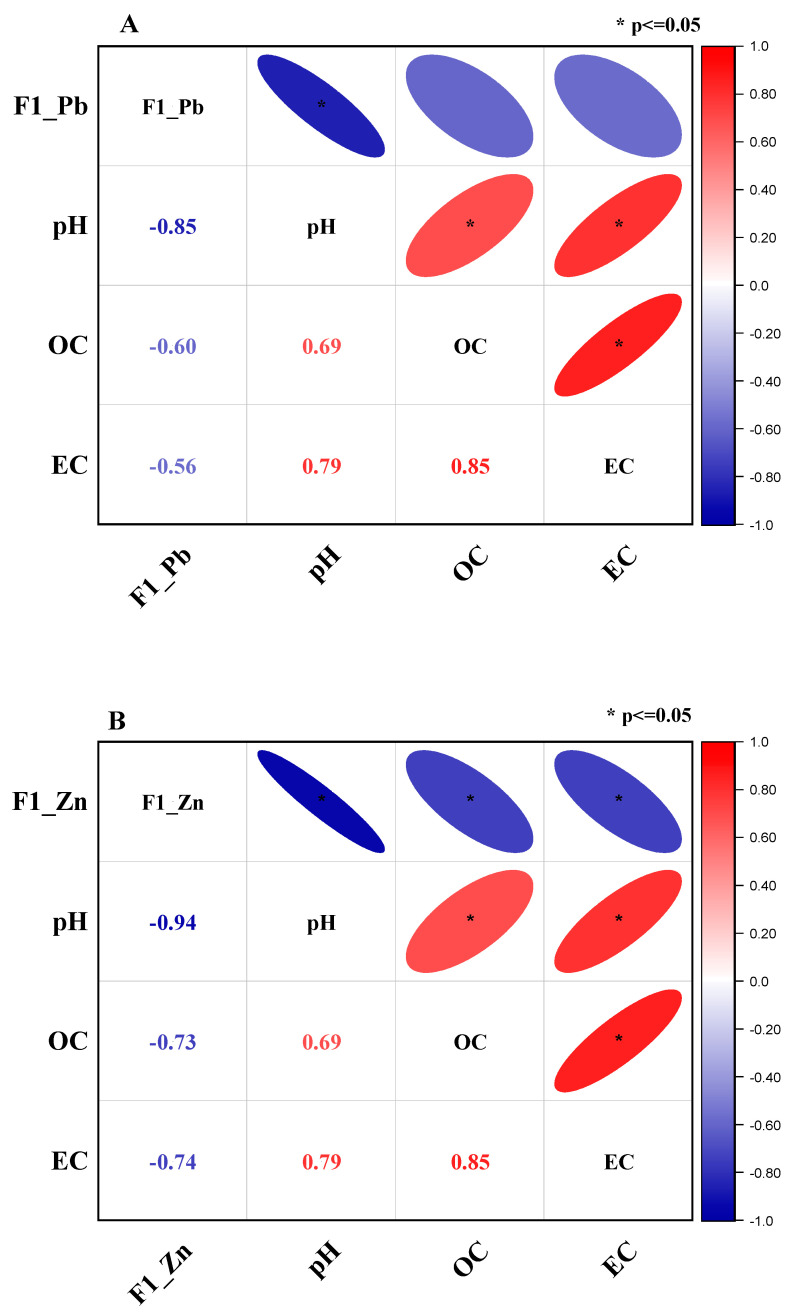
Spearman correlation of exchangeable fraction of Pb (F1–Pb), pH, OC, EC (**A**), and the exchangeable fraction of Zn (F1–Zn), pH, OC, EC (**B**).

**Figure 6 molecules-27-08044-f006:**
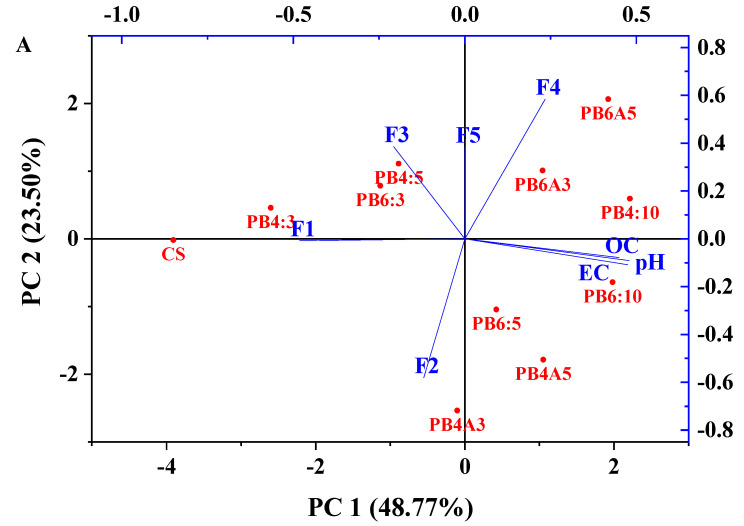
PCA loading plots of F1–Pb, pH, OC and EC (**A**) and F1–Zn, pH, OC and EC (**B**).

**Table 1 molecules-27-08044-t001:** Designation of the incubation experiment.

Sample	Sample Code	Biochar Weight (g)	Apatite Weight (g)	Soil Weight (g)	Ratio (%)
CS	CS	0	0	100	0
CS + 3% PSB400	BC4:3	3	0	97	3
CS + 5% PSB400	BC4:5	5	0	95	5
CS +10% PSB400	BC4:10	10	0	90	10
CS + 3% PSB400 + 3% APA	B4A3	3	3	94	3:3
CS + 5% PSB400 + 5% APA	B4A5	5	5	90	5:5
CS + 3% PSB600	BC6:3	3	0	97	3
CS + 5% PSB600	BC6:5	5	0	95	5
CS +10% PSB600	BC6:10	10	0	90	10
CS + 3% PSB600 + 3% APA	B6A3	3	3	94	3:3
CS + 5% PSB600 + 5% APA	B6A5	5	5	90	5:5

PB: peanut shell biochar; APA: apatite ore; CS: control soil; BC: Biochar; incubation time: 30 days.

**Table 2 molecules-27-08044-t002:** Characteristics of soil and amendments.

Properties	Unit	Soil	PSB400	PSB600	APA
Sand	%	69.78 ± 0.72	na	na	na
Silt	%	5.48 ± 0.32	na	na	na
Clay	%	24.74 ± 0.43	na	na	na
pH		6.47 ± 0.02	10.90 ± 0.01	11.13 ± 0.02	9.36 ± 0.02
OC	%	2.49 ± 0.12	80.79 ± 8.34	73.34 ± 0.21	3.34 ± 0.21
EC	µS cm^−1^	118.50 ± 0.50	>1999	1104.51 ± 1.50	1104.53 ± 1.51
Pb	mg kg^−1^	2447.80 ± 98.60	<LOD	<LOD	<LOD
Zn	mg kg^−1^	2034.30 ± 35.40	0.70 ± 0.03	9.43 ± 0.03	9.43 ± 0.03
Cd	mg kg^−1^	14.10 ± 0.90	<LOD	< LOD	<LOD
S(BET)	m^2^ g^−1^	1.48	79.62	0.45	0.45

Note: mean ± SD; *n* = 3; na: no analysis; LOD: limit of detection; PSB400: Peanut shell-derived biochar produced at 400 °C; PSB600: Peanut shell-derived biochar produced at 600 °C; APA: apatite ore.

**Table 3 molecules-27-08044-t003:** Data analysis of soil organic carbon (OC), pH, soil electrical conductivity (EC) and the exchangeable fraction of Pb (F1_Pb) and Zn (F1_Zn) after 30 days of incubation with biochar and apatite.

Sample	F1_Pb (mg kg^−1^)	F1_Zn (mg kg^−1^)	pH	OC (g kg^−1^)	EC (µS cm^−1^)
CS	495.77 ± 5.20 ^a^	424.82 ± 4.69 ^a^	6.57 ± 0.01 ^h^	19.46 ± 2.11 ^g^	136.4 ± 2.5 ^h^
PB4:3	455.73 ± 15.24 ^b^	416.97 ± 6.93 ^ab^	6.79 ± 0.01 ^f^	26.37 ± 1.21 ^f^	154.8 ± 1.5 ^g^
PB4:5	328.46 ± 7.32 ^c^	348.97 ± 34.08 ^cd^	6.82 ± 0.01 ^e^	42.12 ± 0.65 ^c^	185.3 ± 1.6 ^d^
PB4:10	275.15 ± 31.29 ^d^	277.69 ± 16.52 ^e^	7.24 ± 0.01 ^a^	86.46 ± 0.80 ^a^	211.3 ± 2.3 ^b^
PB4A3	324.88 ± 16.60 ^c^	326.84 ± 23.90 ^d^	7.02 ± 0.01 ^cd^	25.93 ± 1.23 ^f^	190.4 ± 3.2 ^c^
PB4A5	275.37 ± 9.01 ^d^	279.72 ± 32.17 ^de^	7.10 ± 0.01 ^b^	37.73 ± 3.21 ^d^	210.6 ± 5.1 ^b^
PB6:3	319.82 ± 39.99 ^cd^	388.75 ± 27.62 ^bc^	6.67 ± 0.01 ^g^	31.61 ± 1.45 ^e^	175.6 ± 2.5 ^e^
PB6:5	295.26 ± 12.05 ^d^	321.33 ± 22.01 ^d^	6.99 ± 0.01 ^d^	38.63 ± 2.02 ^d^	195.6 ± 1.3 ^c^
PB6:10	234.55 ± 18.27 ^e^	302.89 ± 22.68 ^de^	7.23 ± 0.01 ^a^	77.85 ± 1.04 ^b^	231.3 ± 2.1 ^a^
PB6A3	256.83 ± 28.50 ^de^	308.82 ± 20.76 ^de^	7.05 ± 0.02 ^c^	31.74 ± 1.23 ^e^	183.9 ± 2.4 ^e^
PB6A5	252.83 ± 17.81 ^e^	311.78 ± 11.83 ^d^	7.11 ± 0.01 ^b^	34.65 ± 1.65 ^d^	213.5 ± 3.5 ^b^

Notes: Mean ± SD, n =3; Means followed by the same letters (a,b,c,d,e,f,g,h) within the same column are not significantly different at a 5% level probability.

**Table 4 molecules-27-08044-t004:** Component matrix of the PCA analysis of exchangeable fraction (F1_Pb) of Pb and Zn (F1_Zn) with other factors (OC, pH, and EC) in studied samples after incubation.

Metal	Element	Component 1	Component 2
Pb	F1	−0.48	0.00
	F2	−0.12	−0.58
	F3	−0.21	0.39
	F4	0.23	0.58
	F5	0.23	0.58
	pH	0.47	−0.11
	OC	0.45	−0.08
	EC	0.48	−0.09
	Eigenvalue	3.90	1.88
	Cumulative variances (%)	48.77	72.27
Zn	F1	−0.43	0.00
	F2	−0.23	−0.57
	F3	−0.27	0.66
	F4	0.29	−0.17
	F5	0.27	−0.28
	pH	0.45	−0.06
	OC	0.40	0.32
	EC	0.41	0.17
	Eigenvalue	4.81	1.29
	Cumulative variances (%)	60.10	76.21

## Data Availability

Not applicable.

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
