# Peer review of "Chemical Fractionations of Lead and Zinc in the Contaminated Soil Amended with the Blended Biochar/Apatite"

_molecules, 2022, doi:10.3390/molecules27228044_

Round 1
Author Response
|
Comment #1: Add brief methodology in the abstract (before results) |
Thank you very much for your valuable comment. We added some information of methodoly in the abstract (Line 28-31) |
|
Comment #2: L51-52: use the word “ heavy metals” instead of the pronoun “They” |
We edited that word in (line 59-60) |
|
Comment #3: L65: Add two lines on sources/types of biochar |
Thank you very much for your valuable comment. We added sources/types of biochar (line72-73) |
|
Comment #4: L87: Replace the word “ paper” with “study” |
Thank you for your constructive comment. We edited that word in (line 108) |
|
Comment #5: Entire section of the Methodology should be transferred before the results and discussion. |
We did re-arranged the section of the Methodolody and transferred it before the results and discussion section. (line 124) |
|
Comment #6: What was the size sample of the soil surface samples |
We added the information describing the size of the soil surface samples. (line 562 and 564-565) |
|
Comment #7: L113: use full form first, then the abbreviation |
We used the full form first in the Methodology, then the abreviation for OC and EC. (line 154, 157) |
|
Comment #8: L248-249: overall, the PSB_600 has a large surface area and porosity than PSB_400 and APA, 248 yet fewer organic functional group than PSB_400. Give reasons
|
We added the reasons causing the differences in surface area and organic functional groups of PSB400 and PSB600. (line 333-339) |
|
Comment #9: L404-408: The results of previous studies also showed that biochars promoted the transformation of exchangeable heavy metals to relatively stable forms. Add a few more citations or relevant studies, as stated
|
We did add two more references (line 538) |
|
Comment #10: L434-435: Overall, F1_Pb and F1_Zn had strongly negative correlations with pH, EC and OC, and the correlations between pH and OC, pH and EC, OC and EC were strongly positive. Give reasons
|
Thank you very much for your constructive comment! We added some information to explain the reason why these factors had negative and possitive correlations. (line 563 -578) |
|
Comment #11: Add a few key recommendations based on your finding. - Spelling or sentence correction |
-A few key recommendations based on the present study were added. (line 630-646) -We checked spelling errors and corrected the spelling errors and sentences. (line 353-354, 369, 387, 415, 421, 466, 486, 521, 554, 591-592, 621, 627)
|
|
Comment #12: Rewrite and remove mistakes from the following lines of the manuscript: Inaddition (line 30); was the order (line 32); was obtained (line 394); a significant possibility (line 539). Avoid using pronouns (I/we) in the entire manuscript (line 87) |
We did edited all those flaws. (line 84, 85, 87, 112-113, 153, 250,261, 277, 281, 421-422, 437, 354, 482, 488, 533, 553, 588, 623)
|

Reviewer 2 Report
1. I suggest adding a list of abbreviations.
2. Abstract: I suggest adding some numerical data.
3. Why the peanut shell biochar was selected for the research?
4. Line 86: “The combination of peanut shells derived from biochar and apatite …” is that correct?
5. Subsection 2.1. : Does this subsection discusses the results in Table 1 or in Table 3? The values reported in the body of the subsection are not consistent with those in Table 1. This is a bit inconsistent.
6. Table 1: I suggest to add a legend.
7. Line 122: I suggest to compare with world standards.
8. Line 146, Table 3: I suggest giving detection limits as numerical values.
9. Line 147: Does it follow from Table 1?
10. Line 167 and 178: quazt or quartz? Please check the entire manuscript
11. Materials and methods: Were soil samples collected according to any plan? at what distance were the sampling points located? What was the weight of a single sample? What was the uncertainty of the analytical methods? How was the validity of the results confirmed?
12. I suggest standardizing the way of writing PSB 400 or PSB_400 (similarly for PSB 600)
13. I suggest standardizing the way of recording units
14. Result and discussion: These section just summarized data of published papers without critical analysis. Please add more critical discussion on the above points, in particular with regard to the effectiveness of the method compared to other remediation methods
15. Conclusions: What are the advantages and disadvantages of this method? What could be the problems and limitations in applying this method on a large scale? What are the future perspective for the presented research?
16. Tables: dot or no in the end of title? Please unify
17. Please unify the references list

Author Response
|
Reviewer # 3: |
|
|
Comment #1: I suggest adding a list of abbreviations. |
We added a list of abbreviation. |
|
Comment #2: Abstract: I suggest adding some numerical data.
|
We did add some numerical data as suggested (line 33, 39, 40) |
|
Comment #3: Why the peanut shell biochar was selected for the research? |
We added some information to make clear the reason why peanut shell biochar was selected for the research. (line 103-107) |
|
Comment #4: Line 86: “ the combination of peanut shells derived from biochar and apatite…” is that correct? |
We rewrote the sentence to make it understandable (line 107) |
|
Comment #5: Subsection 2.1.: Does this subsection discusses the results in Table 1 or in Table 3? The values reported in the body of the subsection are not consistent with those in Table 1. This is a bit inconsistent. |
At the beginning, we arranged the structure of the article as the following order: introduction, methods and experiments, results and discussion, conclusion and reference. However, the template of the journal suggested other order as : introduction, results and discussion, methods and experiments, conclusion and reference. Therefore, there was a mismatch of text content and Table. We checked and edited these flaws. (Line 190,217,229) |
|
Comment #6: I suggest adding a legend in Table 1. |
We did add a legend. The abbreviations of samples were described in the Methodology section. (Line 212). |
|
Comment #7: Line 122: I suggest comparing it with the world standard. |
Thank you for your valuable comment! The geological properties of the soil vary among various countries, therefore the limit standards of heavy metals in the argicultural soil vary among different areas and conntries . Moreover, the previous studies used the Vietnamese standard to investigate the polluted level of heavy metals in this area as well (Dang et al., 2019; Vuong et al., 2022). So we would like to choose only the Vietnamese standard of heavy metals for the agricultural soil for this study to discuss the results. However, we did add the heavy metals standard of US EPA 2010 in the text as suggested. (line 201-203) 1. Dang, V.M. et al. (2019) ‘Immobilization of heavy metals in contaminated soil after mining activity by using biochar and other industrial by-products: the significant role of minerals on the biochar surfaces’, Environmental Technology (United Kingdom), 40(24), pp. 3200–3215. Available at: https://doi.org/10.1080/09593330.2018.1468487.
2. Vuong, X.T. et al. (2022) ‘Speciation and environmental risk assessment of heavy metals in soil from a lead/zinc mining site in Vietnam’, International Journal of Environmental Science and Technology, pp. 1–16.
|
|
Comment #8: Line 146: Table 3: I suggest giving detection limits as numerical values |
We provided information of limit of detection (LOD) and limit of quantity (LOQ) in the Table S2 (See Supplement Information) |
|
Comment #9: Line 147: does it follow from Table 1? |
We checked it and rearranged the text to fit to the Table 1 (Line 229) |
|
Comment #10: Line 167 and 178: quazt or quartz? Please check the entire manuscript |
It is quartz. We checked and edited this error. (line 249, 260) |
|
Comment #11: Materials and methods: were soil samples collected according to any plan? At what distance was the sampling point located? What was the weight of a single sample? What was the uncertainty of the analytical methods? How was the validity of the results confirmed? |
-The information of the weight of soil samples was added in (line 126, 129) -The uncerrtainty of the analytical methods was reported in table S2 ( see Supplement Information) |
|
Comment #12: I suggest standardizing the way of writing PSB400 or PSB_400 (similar for PSB600) |
We checked these flaws and standardized them as PSB400 and PSB600. (line 146, 171, 409-410) |
|
Comment #13: I suggest standardizing the way of recording units |
We checked and standardized them . (line 199, 225, 226, 324-325, 377) |
|
Comment #14: Results and discussion: these sections just summarized data of published papers without critical analysis. Please add more critical discussion on the above points, in particular with regard to the effectiveness of the method compared to other remediation methods |
Thank you very much for your constructive comment! We added more critical discussion in the results and discussion section. (Line 490-526) |
|
Comment #15: Conclusions: What are the advantages and disadvantages of this method? What could be the problems and limitations of applying this method on a large scale? What are the future perspective for the presented research? |
Thank you very much for your helpful comment! We did edited the conclusions (line 629-645) |
|
Comment #16: Tables: dot or no in the end of the title? Please unify |
We checked and unified them with dot at the end of the title of all the tables and figures. (line 146-147, 211-214, 249, 290, 314, 385, 387, 396) |
|
Comment #17: Please unify the references list |
We checked and unified them. (line 663-876) |

Reviewer 3 Report
The presented manuscript used peanut shell-derived biochar and apatite alone or in combination to remediate lead and zinc contaminated soil for 30 days and evaluated the effect of both on the ratio of lead and zinc fractions. This is an interesting work. The language of the article is fluent and the analysis is reasonable. However, there is some confusion in the logical structure of the manuscript, and the presentation specification needs further revision. Detailed comments and suggestions are presented as follows:
1. Please revise the serial number of the conclusion to "4".
2. keywords and line 59. "remove" is inaccurate or uncritical.
3. Lines 75 to 77. This sentence is unclear and confusing. Consider breaking it up or deleting it.
4. The fourth paragraph in the introduction can be merged into the last paragraph. Highlight the significance and purpose of the study.
5. Line 120. Please change "mg.kg-1" to "mg kg-1".
6. The data described in the text of section 2.1 are from Table 3, not from Table 1. In addition, the data in Table 3 are clearly presented, so there is no need to use too much language in the main text to express the soil properties. It is recommended that the relevant expressions be placed in the chapter Materials and Methods after shortening them.
7. Line 135. pH values for PSB400, PSB600 and APA do not match the values in Table 3. pH for PSB600 is shown in the table as 9.36. It is possible that the data for PSB600 and APA are mixed up in Table 3, as the values for the base data for both are the same. Please check it.
8. Line 224. Revise "Figures 3" to "Figure 3".
9. Table 1 should follow immediately after section 2.3. This is because the data in Table 1 are discussed in this section. In addition, please align the unit format of EC in Table 1 with the other indicators by writing μS cm-1. The "PB4:3" used in Table 1 and throughout the text could be mistaken for the ratio between two substances, so I suggest that the naming of the samples could be changed to be more easily understood, e.g., "PB4:3" to "3% PB400".
10. I hope the authors had added some language in section 2.4 to further analyze the passivation effect and its causes when PSB and APA are combined.
11. "11pots" described in row 496 does not match the grouping data in Table 4. I only find 9 groups in Table 4. Please check.
12. Line 503. should be followed by a period after Biochar.
13. Line 508. "ratio of 1:2:5" should be changed to "ratio of 1:2.5".
14. Various abbreviations for biochar peanut shell-derived biochar appear, such as BC400 and BC600 in line 485 and lines 315-316, PSB400 and PSB600 in line 135, and PB400 and PB600 in line 356 and in Tables 1 and 4. please standardize the writing style for easy understanding.
15. Section 3.3. Please give more specific information on microwave digestion of soils, such as the volume of the solution and the duration of digestion.
16. From the text and graphs, the experiment seems not to be set in parallel.
17. Table 3. units of EC and S (BET) should be unified with the form of other indicators, written as "m2 g-1". And the figures in the table should be kept in the valid place or the number of decimal places consistent.
18. Conclusion is a little bit longer, please make it more concise and direct to reflect the main paper's goal.
19. Information in the reference (e.g., volume, issue, page number, or article number) is missing or has errors. For example, references 9,12,21,37,49,52,59. Please check it.
Author Response
|
Reviewer # 2: |
|
|
Comment #1: - Please revise the serial number of the conclusion to “4” |
Thank you for your constructive comment. We edited it (line 629) |
|
Comment #2: Keywords and line 59. ‘Remove” is inaccurate or uncritical |
We deleted it (line 68) |
|
Comment #3: L75-77: This sentence is unclear and confusing. Consider breaking it up or deleting it |
We rewrote the sentence to make it understandable. (line 86-87) |
|
Comment #4: The fourth paragraph in the introduction can be merged into the last paragraph. Highlight the significance and purpose of the study
|
We did merge the fourth graph into the last graph and highlight the significance and the purpose of the study (line103-123)
|
|
Comment #5: L120. Please change “mg.kg-1” to “mg kg-1”
|
We edited it. (line 199, 390) |
|
Comment #6: The data describe in the text of section 2.1 are from table 3, not from Table 1. In addition, the data in Table 3 are clearly presented, so there is no need to use too much language in the main text to express the soil properties. It is recommended that the relevant expression be placed in the chapter Materials and Methods after shortening them. |
Thank you for your constructive comments! We rearranged the section 2 and 3 to fit the text with the content (Line 124, 287). We would like to keep the text which describes the soil properties to show its characteristics clearly to the reader |
|
Comment #7: L135, pH values for PSB400, PSB600 and APA do not match the values in Table 3. pH for PSB600 is shown in the table as 9.36. It is possible that the data for PSB600 and APA are mixed up in Table 3, as the values for the base data for both are the same. Please check it
|
We edited that flaw in the table 2 and the text in (line 211 and 216) |
|
Comment #8: Line 224: revise “Figures 3” to “Figure 3”
|
We edited that flaw. (line 306, 317) |
|
Comment #9: Table 1 should follow immediately after section 2.3. This is because the data in Table 1 are discussed in this section. In addition, please align the unit format of EC in Table 1 with the other indicators by writing µS cm-1. The “PB4:3” used in Table 1 ad throughout the text could be mistaken for the ratio between tow substances, so I suggest that the naming of the samples could be changed to be more easily understood, e.g., “PB4:3” to “3% PB400” |
Thank you very much for your constructive comment! - We checked and rearranged the table and the content to match them, after the section 2 and 3 were exchanged, table 1 turned to table 2 (line 211) - For the unit of EC, we checked, edited and unified them in table 2 (line 211) and in the text (line 225). - For sample code: we described the samples for each code and showed clearly what the PB4:3 sample was in the table 4. Moveover, if we changed the name as 3% PB400 we would have to change all the graphs and space in the graphs might not be enough for the new code, therefore we would like to keep the name of the sample as PB4:3 (Table 1, line 146) |
|
Comment #10: I hope the authors had added some language in section 2.4 to further analyze the passivation effect and its causes when PSB and APA are combined
|
We added some discussion on the cause and the effect of PSB and APS when they are conbined. (line 490-526) |
|
Comment #11: “11 pots” described in row 496 does not match the grouping data in Table 4. I only find 9 groups in Table 4. Please check
|
We checked and edited this error. Two more pots were added in the table (line 146). |
|
Comment #12: Line 503: should be followed by a period after biochar
|
We added the necessary information after biochar. (line 147) |
|
Comment #13: Line 508: “ratio of 1:2:5” should be changed to “ratio of 1:2.5”
|
We corrected it. (line 154) |
|
Comment #14: Various abbreviations for biochar peanut shell-derived biochar appear, such as BC400 and BC 600 in line 485 and line 315- 316, PSB400 and PSB600 inline 135, and PB400 and PB600 in line 356 and in Tables 1 and 4. Please standardize the writing style for easy understanding |
We checked and edited them as PSB400 and PSB600 to be unified. (line 146, 171, 409-410) |
|
Comment #15: section 3.3. Please give more specific information on the microwave digestion of soils, such as the volume of the solution and the duration of digestion. |
We provided more information for the microwwave digestion and added a table of the operating parameters of the microwave digestion system in the Supplement Information (Table S1). The volume of the acid solution was provided (line 298). |
|
Comment #16: From the text and graphs, the experiment seems not to be set in a parallel |
We did check and confirm that the experiments were set in a parallel (the same incubation time, the same treatment and analysis methods) |
|
Comment #17: Table 3. Units of EC and S (BET) should be unified with the form of other indicators, written as “m2 g-1”. And the Figures in the table should be kept in valid space or the number of decimal places consistent. |
We checked and unified all the units in Table 3 and the text. (line 211, 324-325) |
|
Comment #18: The conclusion is a little bit longer, please make it more concise and direct to reflect the main paper’s goal. |
We did alter the conclusion a little bit and highlight the conclusion to reflect the main study’s goal. (line 629-645) |
|
Comment #19: Information in the reference (e.g. volume, issue, page number, or article number) is missing or has errors. For example, references 9, 12, 21, 37, 49, 52, 59. Please check it |
We checked and edited errors. Unfortunately, several references which are published recently and have not yet had information of page number, volume..etc. Therefore, we added information of DOI of those articles. (line 663-876) |

Round 2
Reviewer 1 Report
Satisfied with the revision/ changes according to the comments
Author Response
Thank you for your comments.
Reviewer 2 Report
Line 416: please remove the space between the value and unit "63.66 %"
Line 523: please remove the dot between mg and kg "249 mg.kg-1"
Table 4: Please reconsider the number of significant figures
Author Response
|
Reviewer #2 |
|
|
Comment #1: Line 416: please remove the space between the value and unit "63.66 %" |
Thank you very much for your valuable comment. We edited this flaw (line 416) |
|
Comment #2: Line 523: please remove the dot between mg and kg "249 mg.kg-1" |
We did remove the dot between mg and kg (line 524) |
|
Comment #3: Table 4: Please reconsider the number of significant figures |
Thank you very much for your constructive comment. We unified the number of significant figures. (Table 4, line 592) |

Reviewer 3 Report
The authors have made great efforts to improve the manuscript, and now, in my opinion, is ready for publication.
Author Response
Thank you for your comments.